# Malaria in Southern Venezuela: The hottest hotspot in Latin America

**Maria Eugenia Grillet**[1]*, **Jorge E. Moreno**[2], **Juan V. Hernández-Villena**[1], **Maria F. Vincenti-González**[3], **Oscar Noya**[4,5], **Adriana Tami**[3,6], **Alberto Paniz-Mondolfi**[7,8], **Martin Llewellyn**[9], **Rachel Lowe**[10,11], **Ananías A. Escalante**[12], **Jan E. Conn**[13,14]

**1** Laboratorio de Biología de Vectores y Parásitos, Instituto de Zoología y Ecología Tropical, Facultad de Ciencias, Universidad Central de Venezuela. Caracas, Venezuela, **2** Centro de Investigaciones de Campo "Dr. Francesco Vitanza," Servicio Autónomo Instituto de Altos Estudios "Dr. Arnoldo Gabaldón," MPPS. Tumeremo, Bolívar, Venezuela, **3** Department of Medical Microbiology, University Medical Center Groningen, University of Groningen. Groningen, The Netherlands, **4** Instituto de Medicina Tropical, Facultad de Medicina, Universidad Central de Venezuela. Caracas, Venezuela, **5** Centro para Estudios Sobre Malaria, Instituto de Altos Estudios "Dr. Arnoldo Gabaldón", MPPS. Caracas, Venezuela, **6** Departamento de Parasitología, Facultad de Ciencias de la Salud, Universidad de Carabobo. Valencia, Venezuela, **7** Incubadora Venezolana de la Ciencia-IDB. Barquisimeto, Venezuela, **8** Icahn School of Medicine at Mount Sinai. New York, United States of America, **9** Institute of Biodiversity, Animal Health and Comparative Medicine, University of Glasgow. Glasgow, Scotland, United Kingdom, **10** Centre on Climate Change and Planetary Health, London School of Hygiene and Tropical Medicine. London, United Kingdom, **11** Barcelona Institute for Global Health-ISGlobal. Barcelona, Spain, **12** Institute for Genomics and Evolutionary Medicine, Temple University. Philadelphia, United States of America, **13** Griffin Laboratory, Wadsworth Center, New York State Department of Health. Albany, New York, United States of America, **14** Department of Biomedical Sciences, School of Public Health, University at Albany—State University of New York. Albany, New York, United States of America

\* mariaeugenia.grillet@gmail.com, maria.grillet@ciens.ucv.ve

**Data Availability Statement:** Data of this publication are publicly available online from https://figshare.com/account/home, accession

## Abstract

Malaria elimination in Latin America is becoming an elusive goal. Malaria cases reached a historical ~1 million in 2017 and 2018, with Venezuela contributing 53% and 51% of those cases, respectively. Historically, malaria incidence in southern Venezuela has accounted for most of the country's total number of cases. The efficient deployment of disease prevention measures and prediction of disease spread to new regions requires an in-depth understanding of spatial heterogeneity on malaria transmission dynamics. Herein, we characterized the spatial epidemiology of malaria in southern Venezuela from 2007 through 2017 and described the extent to which malaria distribution has changed country-wide over the recent years. We found that disease transmission was focal and more prevalent in the southeast region of southern Venezuela where two persistent hotspots of *Plasmodium vivax* (76%) and *P. falciparum* (18%) accounted for ~60% of the total number of cases. Such hotspots are linked to deforestation as a consequence of illegal gold mining activities. Incidence has increased nearly tenfold over the last decade, showing an explosive epidemic growth due to a significant lack of disease control programs. Our findings highlight the importance of spatially oriented interventions to contain the ongoing malaria epidemic in Venezuela. This work also provides baseline epidemiological data to assess cross-border malaria dynamics and advocates for innovative control efforts in the Latin American region.

number https://doi.org/10.6084/m9.figshare.
12749426.v1.

**Funding:** Funding was provided by the Council for
Sciences and Humanities Development (Grant
CDCH-PG-0382182011 to MEG and JEM) and by
the National Institutes of Health, USA, Grant
2R01AI110112 to JEC and Grant U19 AI089681 to
AAE. The funders had no role in study design, data
collection and analysis, decision to publish, or
preparation of the manuscript.

**Competing interests:** The authors have declared
that no competing interests exist

## Author summary

Despite major progress in malaria control or elimination in South America in the last
years, this disease is still an important public health concern in all the Amazon basin
countries, particularly during 2017 and 2018, when a substantial malaria incidence (~ one
million) was reported. Most of the rise in cases has been due to increases in malaria trans-
mission in Venezuela. Within this country, populations in the southeastern (Guiana
Shield) region have been at the highest risk of *Plasmodium vivax* and *P. falciparum* infec-
tion. In this study, reported malaria cases between 2007–2017 in this region were analyzed
to stratify the spatial risk of disease and identify the malaria surge's main driver over the
last years. Malaria risk was highly focal and prevalent in 2 consistent disease hotspots that
accounted for most *Plasmodium* transmission in the whole region. Illegal gold-mining
activities seem to drive malaria in these disease pockets and seem critical in malaria's
surge throughout the country. Analyses of the Venezuelan surveillance data showed that
from 2014 onwards, local malaria transmission has reemerged in new areas of the country,
leading to a shift in this disease's epidemiology. Successful control of Venezuela's ongoing
malaria epidemic requires hotspot-targeted control at the national level and regional coor-
dination to avoid cross-border malaria spillover.

## Introduction

In the Americas, malaria control programs face significant obstacles that threaten the global
strategy for malaria elimination set for the region for 2030 [1–3]. Progress made between
2000–2015 towards reducing malaria morbidity and mortality stalled in 2016 and reversed in
2017–2018 [3–5] when cases escalated to ~1 million in each year [2,3]. Since 2014, the Venezu-
elan economy's collapse, and with it, its entire healthcare system, have been a significant driver
for this surge [6,7]. In particular, Venezuela reported a total of 1,255,299 cases between 2015–
2018, the highest number in Latin America, and in 2017 exhibited one of the most substantial
increases in malaria cases worldwide [1,2,8].

As in other endemic countries in South America [5], *Plasmodium vivax* accounts for the
majority of reported cases (76%) in Venezuela followed by *Plasmodium falciparum* (17.7%),
mixed *P.vivax* /*P. falciparum* infections (6%) and *Plasmodium malariae* <1% [3]. Venezuela
successfully achieved malaria elimination in approximately 75% of its territory during the
early 1960s [9]. However, low to moderate malaria transmission by *P. vivax* and *P. falciparum*
persisted in the lowland Amazon rainforests and savannas of the remote Guayana region,
mainly in isolated Amerindians communities, South of the Orinoco River [10]. During the
1980s, *P. vivax* malaria reemerged along the coastal wetlands in the country's northeastern
region [11], but its transmission was interrupted twenty years later [12]. However, malaria
transmission in Bolívar state, a region in southeastern Venezuela bordering Brazil and Guyana,
has persisted. Since 1990, this region has contributed > 60% (1992–1995) to 88% (2000–2014)
of Venezuela's total malaria cases [12,13]. Gold mining has been associated with the high
malaria incidence in specific municipalities of this region where gold miners seem to account
for an estimated 47% to 80% of malaria cases [4,5,13]. Clearing forest for mining activities is
observed to create favorable conditions for the main vector malaria species in this area
[5,7,13]. The health situation in Bolívar state has worsened significantly in recent years, and
current limitations in malaria surveillance and control are a cause of major concern [7,13].

Additionally, since 2014, local malaria transmission has reemerged in previously endemic and new areas across the country, including Venezuela's northeastern region [7].

At present, the uncontrolled upsurge of malaria, coupled with Venezuelans' mass migration to neighboring countries, poses a serious threat to the broader region [6,7,14], jeopardizing efforts in bordering malaria-endemic countries to achieve their goals for disease control. Human population movement among countries included in the Guiana Shield (Guyana, French Guiana, Suriname, and parts of Colombia, Venezuela, and Brazil) serve to amplify the regional impact of malaria increase in the state of Bolívar. In addition, there is a latent risk that a regional malaria corridor will form [15] from Bolívar state to northern Brazil via the mass migration of displaced individuals who take advantage of the existing roads and trails connecting Venezuela and Brazil (called Route 10). This situation is worsened by the presence of novel mutations linked to artemisinin resistance in Guyana [16], raising concerns about the potential regional spread of mutations linked to drug-resistant and artemisinin delayed parasite clearance in *P. falciparum* where Venezuela could act as a hub [17].

Considering all these factors, a first step to better understand how to manage this public health crisis is to characterize malaria transmission's spatiotemporal dynamics in southern Venezuela. Heterogeneity and clustering of cases within an endemic focus (an area with suitable environmental and epidemiological characteristics that sustain transmission) is an emerging pattern common to mosquito borne-infectious diseases [18]. As part of such spatial heterogeneity, there could be hotspots (areas where transmission intensity exceeds the regional average) responsible for most cases in a focus. Such areas play a critical role in malaria's resilience to interventions because they frequently maintain persistent transmission, acting as a source of infection for the entire region [18]. Thus, we analyzed reported (passive surveillance) malaria cases data (*P. vivax* and *P. falciparum*) from 2007 through 2017 in Bolivar state. The study aimed to identify critical hotspots and determine their transient or persistence dynamics, drivers, and role in the malaria surge observed over the last few years. Particularly, we expected that illegal gold mining activities disproportionally contributed to the transmission of both parasites in the detected hotspots, a socioeconomic malaria risk factor that has been suggested in previous studies [7,13,19]. Finally, we described the spatial epidemiology of malaria at a national level and assess the extent to which landscape epidemiology of this infection has changed throughout the 2014–2017 period.

## Method

### Study area

Venezuela, located on the northern coast of South America, has a surface area of ~916,445 km$^2$ (Fig 1A). Bolívar state (Fig 1B), the focus of this study, covers an extension of 240,500 km$^2$ and as of the 2018 Venezuelan census [20], has a population of 1,837,485 inhabitants heterogeneously distributed across 11 municipalities and 46 parishes. This region has a low average population density (5.1 inhabitants per km$^2$) and extensive territory. Population densities per municipality are variable and range from 553 (e.g., Caroni municipality) to < 1 inhabitant per km$^2$ (e.g., Gran Sabana municipality). Most of the population resides in the northernmost and easternmost parts of the state (Fig 1B), where economic activities are related to mining (iron, gold, bauxite, and diamonds), production of steel, aluminum, and hydroelectric industries, businesses and services, forestry, cattle, and agricultural development. The state capital is Ciudad Bolivar (Heres municipality) but Ciudad Guayana (Caroni municipality) is the main urban center. Although these northern cities are well-connected with the rest of Venezuela, the most southern and western rural areas are sparsely populated with few or no connecting roads. There is one major paved road, Route 10, that runs along the eastern side of Bolívar and

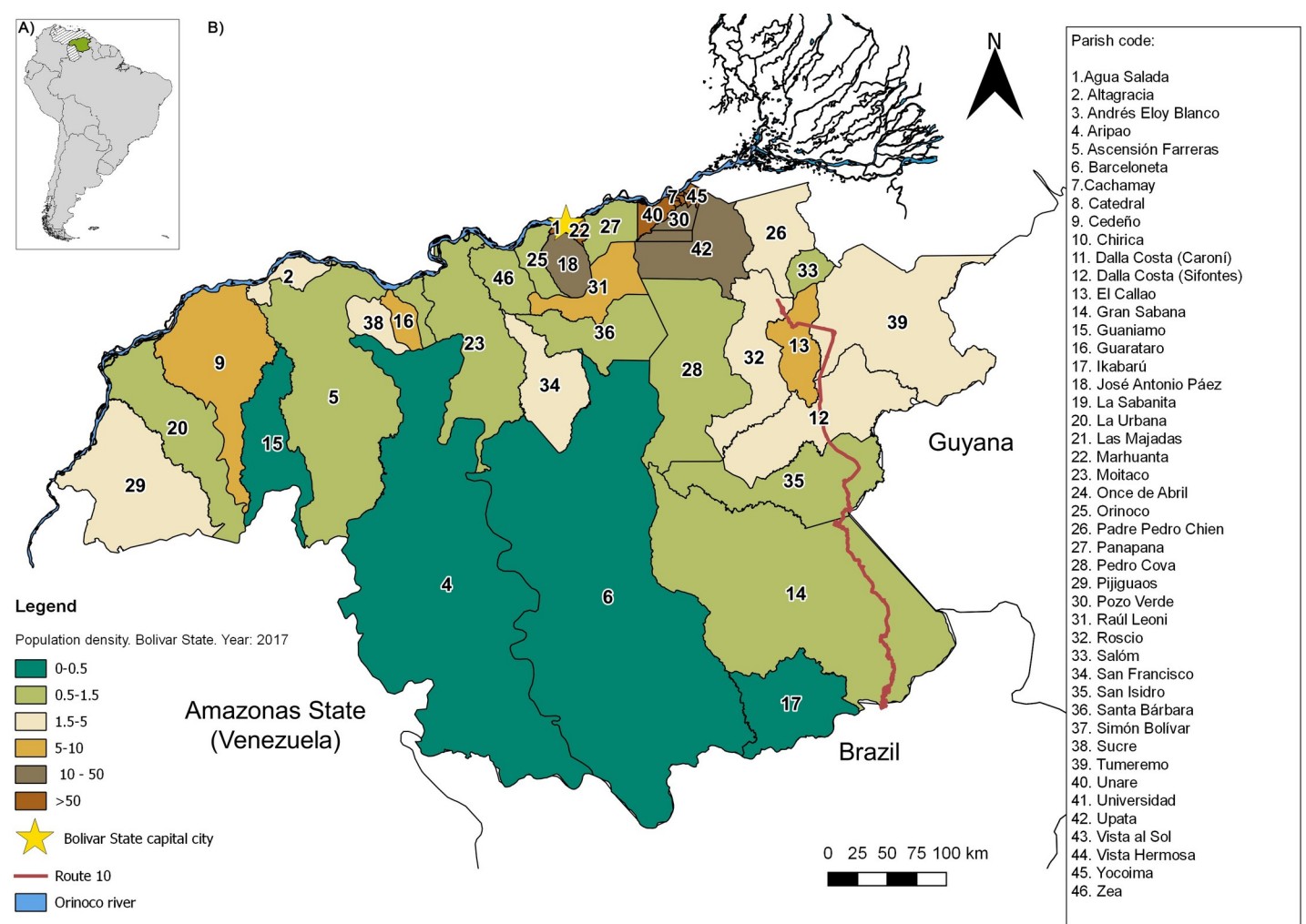

**Fig 1.** A) Map of Venezuela, South America, and the study area (Bolívar state in green). B) Map of Bolívar state showing the population density by civil parish (people per square kilometer) and other relevant features of the landscape. Maps were created with the Q-GIS software (https://www.qgis.org/es/site/).

connects Venezuela with Brazil, crossing through the state's main endemic malaria region (Fig 1B). In this region, malaria does not exhibit a seasonal pattern, particularly for *P. vivax*, which is characterized by a near perennial incidence [12,13]. The primary vectors for all *Plasmodium* species in the state are *Nyssorhynchus darlingi* (also referred to as *Anopheles darlingi*) and *Ny. albitarsis* s.l. (also referred to as *Anopheles albitarsis*) [19,21,22,23].

## Epidemiological and sociodemographic data

The yearly number of *P. vivax* and *P. falciparum* cases per parish (2007–2017) in Bolivar state were provided by the Malaria Control Program, a local branch of the Venezuelan Ministry of Health [24]. Time series of annual malaria incidence (2014–2017) per municipality at the country level were obtained from the PAHO Malaria Surveillance Indicators [25]. Malaria incidence rates per 1,000 inhabitants by parish and by *Plasmodium* species were accounted. These rates were calculated considering the human population growth rate predicted for the studied period according to the demographic data (at-risk population) from the National Statistics Office of Venezuela [20]. In addition, we obtained information (retrospective analysis) from the local

(Bolivar state) malaria database [24] of the demographic and epidemiological profiles of all individuals diagnosed with malaria during the studied period (2007–2017). Particularly, we were looking for whether or not miners were well represented in the occupational category of all the Bolívar state patients. We also examined other demographic factors such as age and sex.

Historically, Venezuela's local health services have reported malaria cases detected by passive surveillance on a weekly basis according to geographic origin and symptomatic cases. Case confirmation usually requires clinical and epidemiologic data (e.g., exposure history in endemic areas) plus a positive blood smear. The Venezuelan Ministry of Health (MofH) uninterruptedly has issued weekly and monthly epidemiological reports known as the "*Boletín Epidemiológico*" reporting on the number of malaria cases, type of infection (parasite species), distribution per state, municipality and parish, distribution per age and sex, occupation of the patient, risk maps, and the number of detected imported cases. All reported malaria cases are notified at the country level via the National Notifiable Diseases Surveillance System (NDSS) by filling out a questionnaire containing relevant epidemiological inquiries. Despite the Venezuela's health-care crisis, local malaria surveillance and data collection continued to occur and weekly bulletins were made available until 2017 but have been interrupted [7].

## Data analyses

We conducted a secondary analysis of passive malaria surveillance data reported by the Venezuelan Minister of Health in Venezuela between 2007 and 2017. Specifically, the annual malaria incidence per parish in Bolivar was mapped to characterize spatio-temporal disease dynamics. Local spatial autocorrelation analysis was performed using Anselin Local Moran's I test to identify malaria's significant spatial patterns [26]. Specifically, we evaluated the likelihood of malaria occurring equally at any location (parish centroid) within Bolivar state vs. the detection of unusual aggregations of malaria incidence. This test evaluates adjacent positions and finds a strong positive spatial autocorrelation when the surrounding incidence of disease over the entire study area has analogous values (called "High-High" areas). A positive value for 'I' indicates that the adjacent parishes are bounded by *P. vivax* or *P. falciparum* incidences with similar values. Such a feature is considered a cluster or hotspot. To test whether the observed clustering/dispersing is statistically significant, an estimated Z-score and a 99% level of significance ($P < 0.001$) were selected.

We also explored occupational risks and demographic factors in hotspot areas by analyzing the proportion of the diagnosed cases during the studied period (2007–2017) that worked primarily in mining, agriculture, forestry, fishing, retail, and other, as well as examining the ratio of male-to-female and age-structure. The chi-squared test allowed us to determine whether there were significant differences between the expected frequencies and the observed frequencies for one or more categories.

To identify illegal mining operations in the study area, we obtained data on the deforested land cover distribution areas in the parishes classified as hotspots across the study. Data on deforestation were obtained from the Global Forest Watch [27] (http://www.globalforestwatch.org) and remote sensing images from Landsat Thematic Mapper imagery [28] (www.http//earthexplorer.usgs.gov/). Illegal gold mining has been the main cause of deforestation in southern Venezuela in recent years [29]. In order to determine if malaria cases exhibited an increasing trend (distinguished from random behavior) along the years or along the temporal cover tree loss, the best fit line (curve fitting approach) to the time series was applied to aid interpretation of data.

Finally, we used kriging, a local geostatistical interpolation method [30], to generate an estimated continuous surface from the scattered set of points (i.e., municipality centroids) with z-

values (i.e., annual parasite incidence in each municipality) to better capture the local spatial variation of malaria spread (spatial risk diffusion map) across the country during the 2014–2017 period. The main assumption of this technique is that the distance or direction between sample points reflects a spatial correlation that can be used to explain the variation of the values of the variable, malaria incidence, on the whole surface. To do that, first, it creates a variogram and covariance function to estimate the statistical dependence (autocorrelation) values. The next step is to fit a model-that, is a continuous function or curve- to the points forming the empirical variogram and predict attribute values (malaria incidence) at unsampled locations. Several variogram models were tested using a cross-validation procedure. The best model was adjusted for any directional spatial trend in our data (anisotropy) in the semi-variogram [30].

Maps, hotspots, descriptive analyses, remote sensing image visualization, and kriging analyses were performed in Q-GIS software (version 2.18.9-Las Palmas de G.C., GNU-General Public License, https://www.qgis.org/es/site/) and the R-software (The R-Development Core Team, http://www.r-project.org).

## Results

The total number of accumulated cases of malaria in Venezuela during the sampled decade was 1,207,348 (range: 32,037–411,586), with overall malaria incidence rates (cases/1,000 inhabitants-year) increasing from 5.2 (2007) to 28 (2017; Fig 2). During this period malaria incidence increased nearly 10-fold (2007: 41,749 cases; 2017: 411,586) with a sharp and significant increase since 2014 (Exponential model, $R^2 = 0.92$, $P < 0.05$). Due to malaria infection, the overall mortality increased nearly 20-fold from 16 (2007) to 312 cases in 2017. Bolívar contributed with ~47% of the total cases in Venezuela during 2017; but during the previous years, this region accounted for 60 to 80% of the reported malaria for the whole country (Fig 2). In

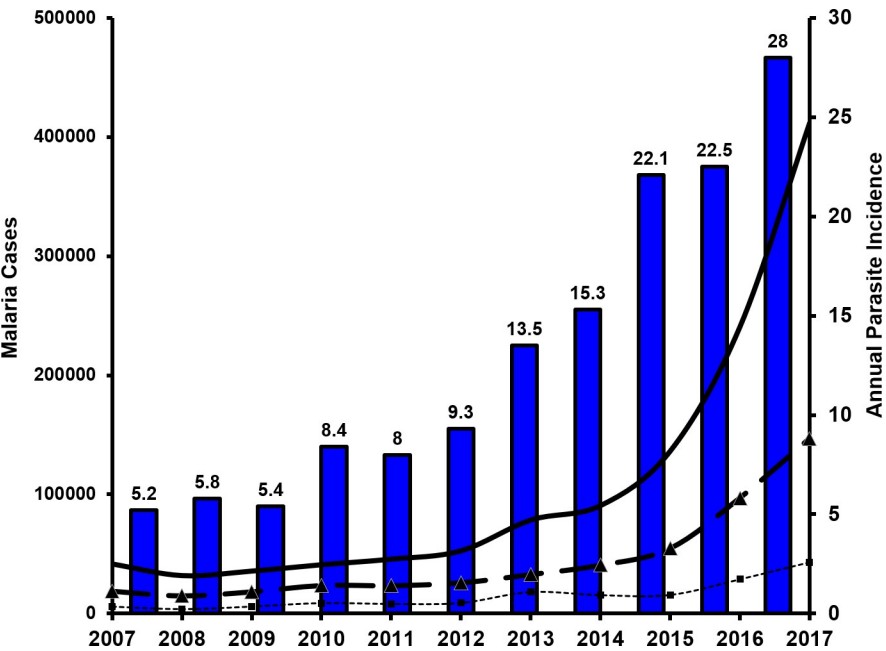

**Fig 2. Number of confirmed malaria cases in Venezuela (solid line) and Bolívar state (dashed line: *P. vivax* and small dashed line: *P. falciparum*).** Annual parasite incidence (API: Number of confirmed malaria cases/1,000 inhabitants, blue bars) in Venezuela (2007–2017).

Bolívar, 20–30% of malaria cases were due to *P. falciparum*, with the remaining 70–80% attributed to *P. vivax*, with a mean ratio of *P. vivax*/*P.falciparum* of 3.04 (+/- 0.17 SE). Fig 2 also reveals that the malaria burden due to *P.vivax* in Bolívar State has had a similar and steady 4.5-fold increment ($R^2$ = 0.93, $P$ < 0.05) since 2014 (from 32,791 to 146,885 cases) with an accumulated 496,847 cases for that time period. A more moderate increase (~2.5-fold, $R^2$ = 0.89, $P$ < 0.05) was detected for malaria morbidity due to *P. falciparum* (18,127 cases in 2014 to 43,152 in 2017), with 152,589 accumulated cases for the same period (Fig 2).

In Bolívar state, annual *P. vivax* incidence during 2007–2017 (Figs 3A and 3B and S1) was heterogeneously distributed, with most cases spatially concentrated in the mid-eastern and southern parishes of San Isidro, Dalla Costa and Ikabarú (the latter bordering Brazil), followed by northern Pedro Cova, Santa Barbara, and El Callao Parishes. Few or no cases were reported in the remaining parishes. Parishes with the presence of malaria cases had annual *P. vivax* incidences as high as 4,672 cases/1,000 inhabitants (2017) and mean annual incidences of up to 1,919 cases/1,000 inhabitants-year (e.g., San Isidro). The temporal dynamics of *P. vivax* cases in San Isidro were representative of the dynamics of malaria overall in the state Bolívar accounting for ~43.4% of the overall malaria morbidity during the study period. As with *P. vivax*, there was also substantial spatial heterogeneity for *P. falciparum* (Figs 3C and 3D and S2). This parasite species had the highest annual incidence with 1,549 cases/1,000 inhabitants (2017) and the highest annual mean of 738 cases /1,000 inhabitants (2017) both reported for San Isidro Parish. Further spatial analyses revealed significant local clustering of *P. vivax* and *P. falciparum* incidences between the two parishes of San Isidro and Dalla Costa, providing additional evidence that malaria transmission has focused and persisted in these localities

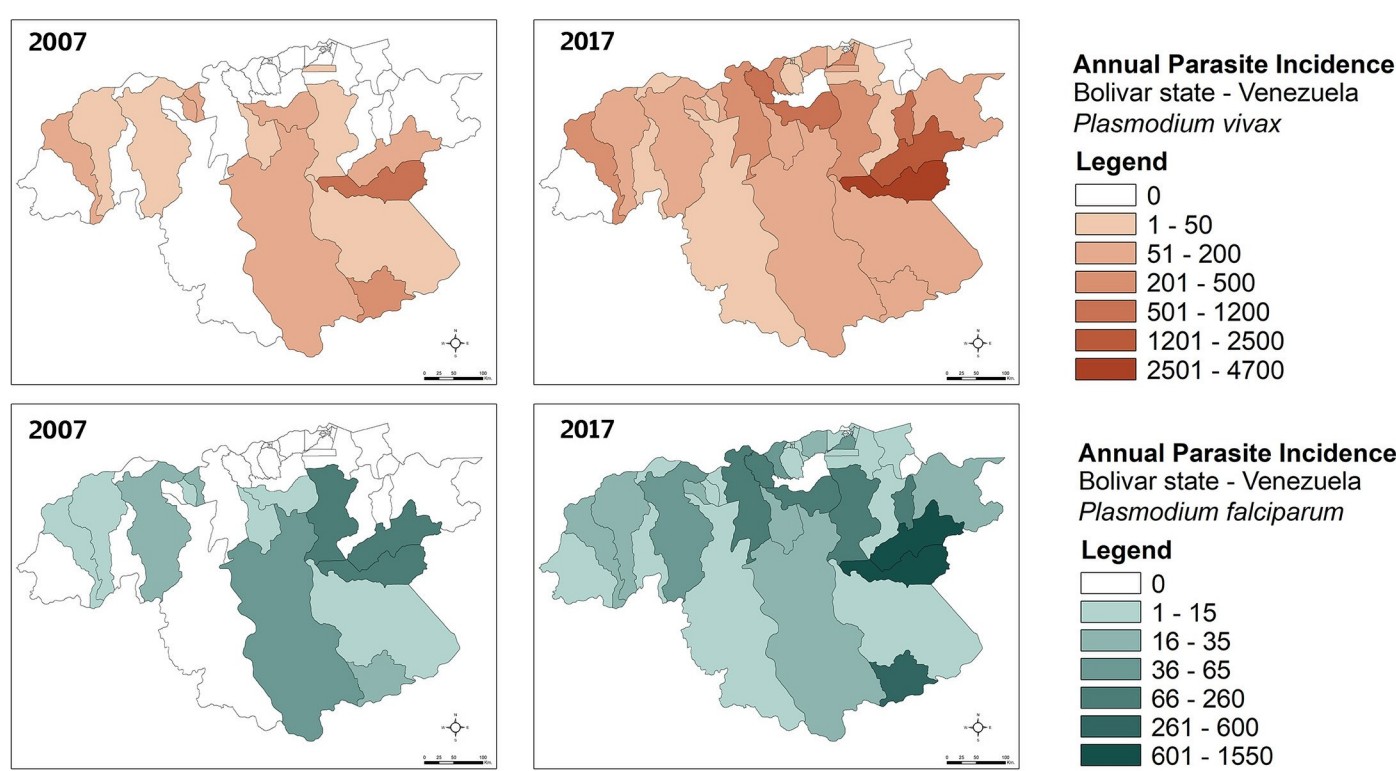

**Fig 3.** Annual *P. vivax* (A, B) and *P. falciparum* (C, D) incidence (API: Number of confirmed malaria cases/1,000 inhabitants) per parish in Bolívar state (2007 and 2017), south-eastern Venezuela. Maps were created with the Q-GIS software (https://www.qgis.org/es/site/).

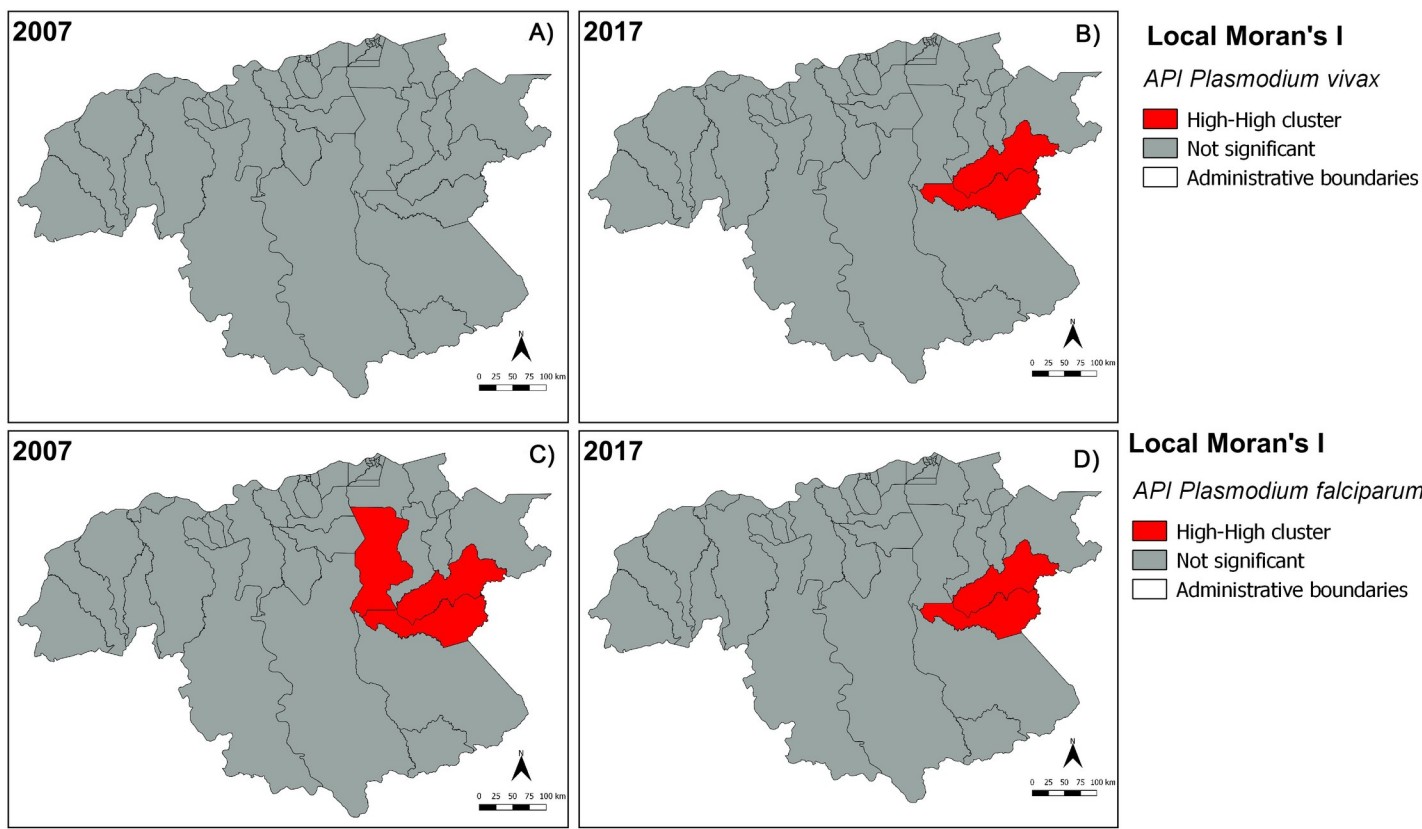

**Fig 4.** Significant clusters (hotspots) of annual *Plasmodium vivax* (A, B) and *P. falciparum* (C, D) incidence (red-red) in Bolívar state (south-eastern Venezuela), during 2007 and 2017. Maps were created with the Q-GIS software (https://www.qgis.org/es/site/).

supporting their 'hotspot' status within the state of Bolívar over the last decade (Figs 4 and S3 and S4).

In San Isidro, throughout the study period *P. vivax* cases predominantly occurred in men (70%: 145,492/206,455, $X^2$ = 34,608.48, 1df, $P$ < 0.0001, Table 1), mainly affecting age-groups between 21–30 (32%), 11–20 (22%) and 31–40 (20%) years (Table 1 and S5 Fig). From an occupational perspective, most of the individuals who were infected with *P. vivax* during the study period were linked to gold mining activities (miners) compared with other common occupations including housekeeper, machine operator, and student, among others (62%: 127,898/206,455; range: 56%-68%, $X^2$ = 11,792.08, 1df, $P$ < 0.0001, Table 2). A similar significant association was found for *P. falciparum*: gold miners accounted for ~ 66% of cases (50,013/75,124; range: 63%-70%, $X^2$ = 8,254.48, 1df, $P$ < 0.0001, Table 2).

The concentration of malaria cases in southern San Isidro coincided with deforestation areas as a result of illegal mining activities. During the study period, most villages from Sifontes Municipality that had >1,000 cases (San Isidro and Dalla Costa parishes) were located in this open (deforested) area (Fig 5A), where the percentage of deforestation (tree cover loss) has dramatically increased in recent years (Fig 5B). Since 2007, San Isidro concurrently lost 3,058 Hectares tree cover (~1.02% decrease) while malaria increased by ~746% (Fig 6A). Similarly, Ikabaru Parish (municipality of Gran Sabana), lost 2,934 Hectares of tree cover (~1.06% decrease) and registered a significant and paralleled increase in malaria cases (Fig 6B).

Fig 7 depicts both the spatial spread of malaria (annual parasite incidence) across Venezuela that has expanded from southern Guayana toward the northern-central-western areas during

**Table 1. Demographic characteristics of malaria patients[1] from two parishes of the Bolivar state (South-eastern Venezuela), during 2007–2017 period.**

| Species | P. vivax, n (%) | | P. falciparum, n (%) | |
| --- | --- | --- | --- | --- |
| Parish | San Isidro | Dalla Costa | San Isidro | Dalla Costa |
| Total Cases | 206,455 | 77,896 | 75, 124 | 30,291 |
| **Age (years)** | | | | |
| < 10 | 16,705 (8.1) | 6,313 (8.1) | 4,004 (5.4) | 1,499 (5.0) |
| 11–20 | **44,445 (21.5)**[*] | **15,699 (20.1)**[*] | **14,003 (18.6)**[*] | **5,079 (16.8)**[*] |
| 21–30 | **66,639 (32.3)**[*] | **2,5371 (32.6)**[*] | **24,423 (32.5)**[*] | **9,905 (32.7)**[*] |
| 31–40 | **41,629 (20.2)**[*] | **15,905 (20.4)**[*] | **16,604 (22.1)**[*] | **7,028 (23.2)**[*] |
| 41–50 | 23,087 (11.2) | 9,008 (11.6) | 10,020 (13.3) | 4,173 (13.8) |
| 51–60 | 10,337 (5.0) | 4,110 (5.3) | 4,619 (6.1) | 1,911 (6.3) |
| 61–70 | 2,802 (1.3) | 1,160 (1.5) | 1,173 (1.6) | 569 (1.9) |
| 71–80 | 610 (0.3) | 268 (0.3) | 221 (0.3) | 106 (0.3) |
| 81–90 | 120 (0.06) | 47 (0.06) | 46 (0.06) | 17 (0.06) |
| > 90 | 81 (0.04) | 15 (0.02) | 11 (0.02) | 4 (0.02) |
| **Sex** | | | | |
| Male | **145,492 (70.5)**[*] | **55,065 (70.7)**[*] | **54, 338 (72.3)**[*] | **21,620 (71.4)**[*] |
| Female | 60,963 (29.5) | 22,831 (29.3) | 20,786 (27.7) | 8,670 (28.6) |

[1] Values in the table (percentage) correspond to attributes of malaria-confirmed cases from the Malaria Control Program (Minister of Health) database of the two main malaria hotspots (Bolivar state, South-eastern Venezuela) during the 2007–2017 period.

[*]Bold figures indicate that malaria cases occurred predominantly ($X^2$-test, 1df, $P < 0.0001$) in men, and highlighted age-groups

(see the main text for details and better explanation, including S5 Fig).

**Table 2. Occupational categories of malaria patients[1] from the two parishes (Bolivar state, South-eastern Venezuela) where disease hotspots were detected (2007–2017).**

| Species | P. vivax, n (%) | | P. falciparum, n (%) | |
| --- | --- | --- | --- | --- |
| Parish | San Isidro | Dalla Costa | San Isidro | Dalla Costa |
| **Categories** | | | | |
| Gold Miner | **127,898 (61.9)**[*] | **48,051 (61.7)**[*] | **50,013 (66.6)**[*] | **19,984 (65.9)**[*] |
| Cook | 137 (0.07) | 49 (0.06) | 48 (0.06) | 19 (0.06) |
| Student | 17,901 (8.7) | 5,236 (6.7) | 5,031 (6.7) | 1,447 (4.8) |
| Housekeeper | 44,380 (21.5) | 16,541 (21.2) | 16,076 (21.4) | 6,706 (22.1) |
| Worker | 2,260 (1.1) | 555 (0.7) | 403 (0.5) | 115 (0.4) |
| Machine Operator | 7,829 (3.8) | 4,168 (5.3) | 1,935 (2.6) | 1,035 (3.4) |
| Fisherman | 58 (0.03) | 39 (0.05) | 12 (0.02) | 7 (0.02) |
| Retailer | 497 (0.2) | 204 (0.3) | 85 (0.1) | 30 (0.09) |
| College Professor | 95 (0.05) | 72 (0.09) | 21 (0.03) | 21 (0.07) |
| School Teacher | 2 (0.001) | 7 (0.01) | 1 (0.001) | 0 (0.0) |
| Military | 25 (0.01) | 41 (0.05) | 4 (0.005) | 12 (0.04) |
| Unemployed | 104 (0.05) | 34 (0.04) | 16 (0.02) | 6 (0.02) |
| Private Employee | 121 (0.06) | 42 (0.05) | 16 (0.02) | 5 (0.02) |
| Public Employee | 313 (0.1) | 163 (0.2) | 95 (0.1) | 53 (0.2) |
| Agriculture | 889 (0.4) | 468 (0.6) | 195 (0.3) | 130 (0.4) |
| Merchant | 904 (0.4) | 330 (0.4) | 254 (0.3) | 122 (0.4) |
| Others | 3,042 (1.5) | 1,896 (2.4) | 919 (1.2) | 599 (1.9) |
| Total Malaria Cases | 206,455 | 77,896 | 75,124 | 30,291 |

[1] Values in the table (percentage) correspond to attributes of malaria-confirmed cases from the Malaria Control Program (Minister of Health) database of the two main malaria hotspots (Bolivar state, South-eastern Venezuela) during 2007–2017 period.

[*]Bold figures indicate that malaria cases occurred predominantly ($X^2$-test, 1df, $P < 0.0001$) in miners (see the main text for details).

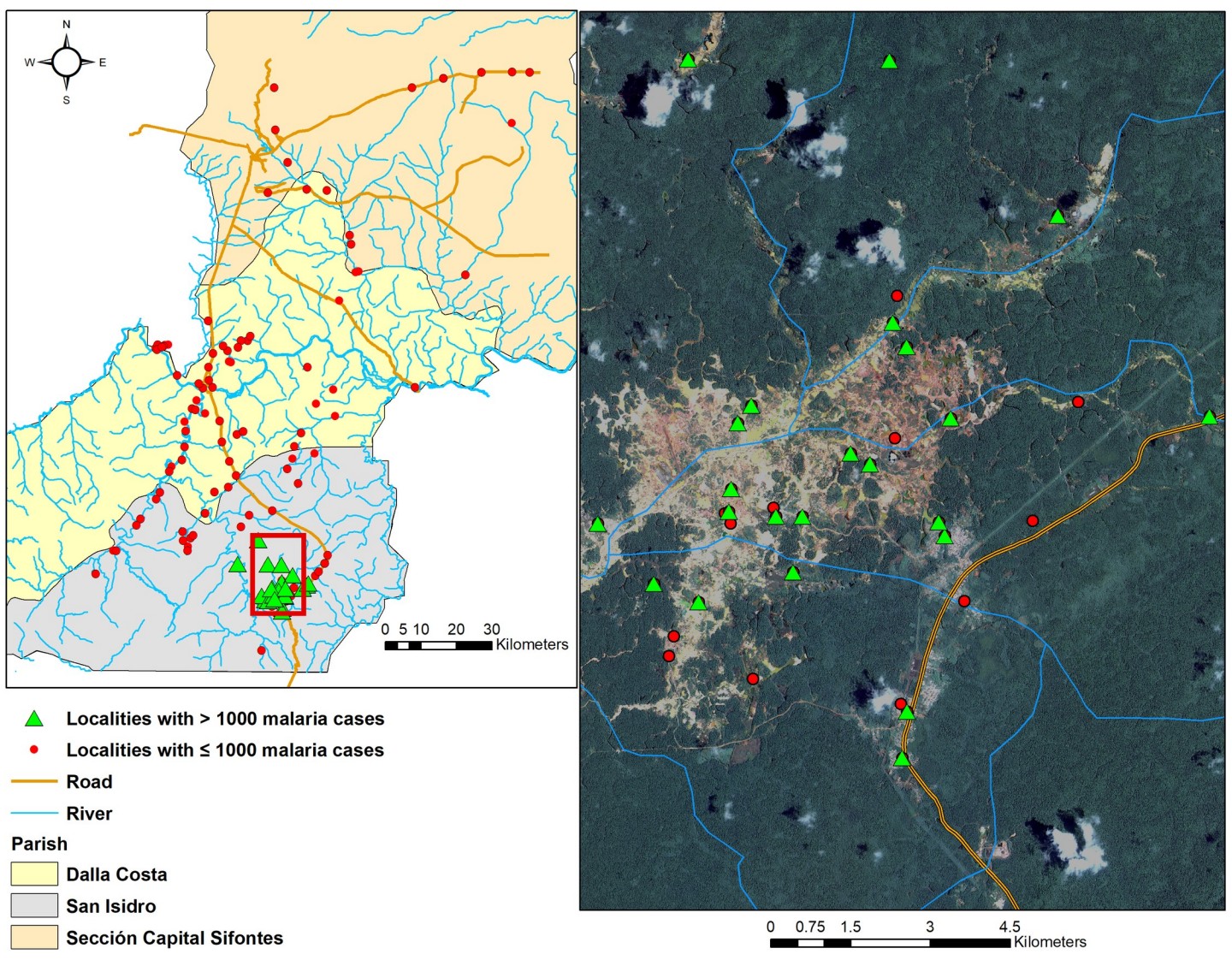

**Fig 5.** Left: Sifontes Municipality and its three parishes showing the spatial distribution of all localities and some general landscape features. Right: satellite image (source: Landsat 8, spatial resolution: 15m, date: 12/19/2017) of "Las Claritas" (framed in red window in San Isidro Parish); green triangles: localities with > 1,000 malaria cases across the study period in the deforested mining area; red dots are localities with < 1,000 malaria cases. Maps were created with the Q-GIS software (https://www.qgis.org/es/site/).

2014–2017 (Epidemiological week-EW 21) and the intensification of disease transmission in the South, an endemic area of sustained concern. Particularly, this spatial prediction analysis emphasizes that the primary high-risk malaria areas and potential sources of parasite dispersal within the country are the hotpots present in the state of Bolívar, followed by the southwestern state of Amazonas. Considering the population growth during that period, the national percentage of individuals living in areas at risk of contracting malaria increased from 34.4% (9,907,708 people) to 50% (15,988,534 people) between 2014 and 2017.

## Discussion

The risk for malaria in southeastern Venezuela varied widely with most cases reported in the mid-eastern and southern parishes of the state of Bolívar, where we identified two persistent

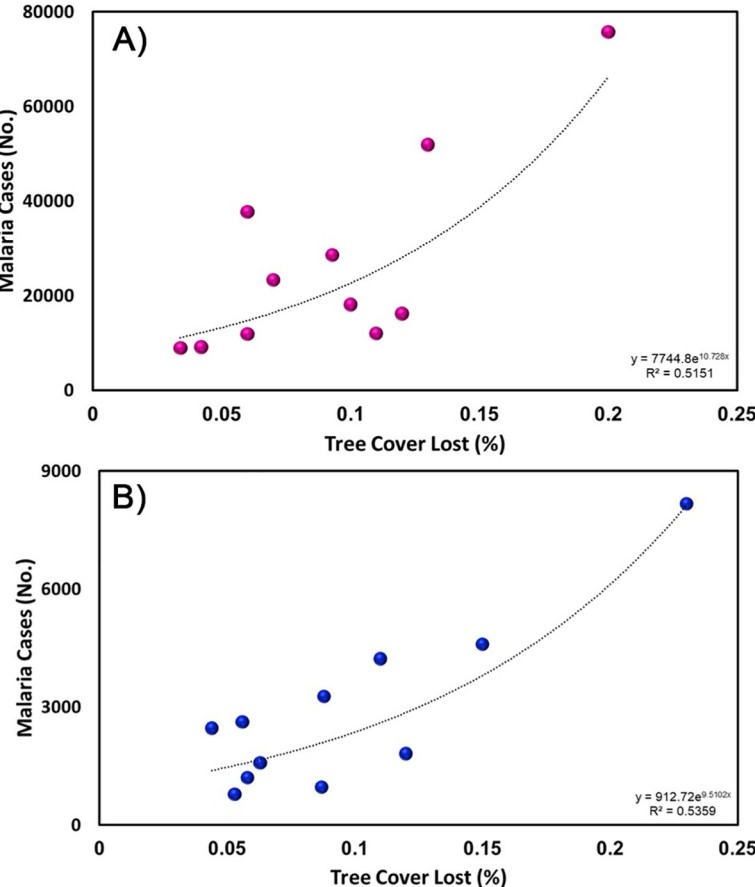

**Fig 6.** Percentage of deforestation (tree cover lost) and accumulated malaria (*P. vivax* + *P. falciparum* cases) in Sifontes Municipality (A) and Gran Sabana Municipality (B) in Bolívar State.

hotspots. Local transmission from these infectious disease pockets accounted for highest malaria transmission in the entire region (~ 61%) and country (> 60%) over time (2007–2017). Both hotspots have been a long-standing regional source of *P. vivax* and *P. falciparum* transmission, as suggested by earlier studies [7,13,21]. Our results support previous findings from Venezuela, Brazil and Peru showing that *P. vivax* malaria spatial heterogeneity is characterized by high-risk localities interspersed with others showing low to moderate risk [11,31–33]. Due to their stability and / or persistence over time, these well characterized hotspots could be predictive of prospective malaria incidence in the surrounding areas as has been found in similar studies [34–36].

Our results highlight that disease patterns at larger spatial scales are driven by a sum of factors acting at local scales [37], such as mosquito ecology (especially larval habitats and host-seeking behavior) and at-risk human population dynamics (e.g., density, distribution, and mobility). In particular, our findings support the hypothesis that illegal gold mining is one of the leading local socioeconomic drivers for malaria in southeastern Venezuela and a major contributing factor to the upsurge of malaria in the recent years. First, we found that areas deforested by gold mining had more clusters of *Plasmodium* cases compared to areas nearby. Secondly, both *P. vivax* and *P. falciparum* increased in incidence (4-8-fold) over time in those regions along with a concomitant decrease of vegetation cover (3-6-fold) resulted from such mining activities. Finally, our results reporting age and sex patterns within the region's malaria

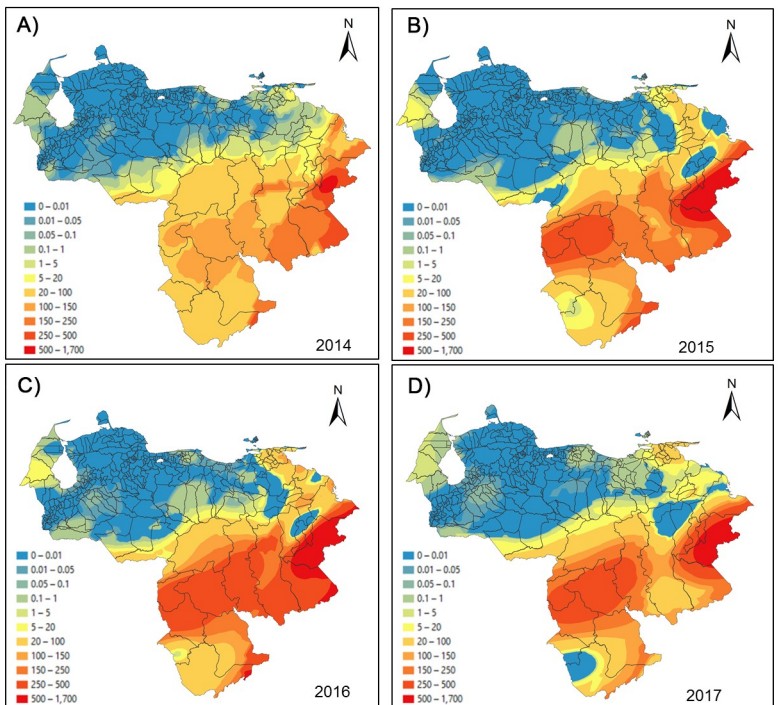

**Fig 7.** Spatial prediction maps (malaria risk) for 2014 (A), 2015 (B), 2016 (C) and 2017 (D) years derived from the ordinary gaussian kriging interpolation model of the annual parasite incidence (API). Note Year 2017, data available only through Epidemiology Week 21. Maps were created with the ArcGIS desktop software (https://www.esri.com).

patients are compatible with the occupation as a factor determining the greater and prolonged exposure to vector bites of young males in the mining fields compared to girls and women.

Illegal gold mining and the associated deforestation have rapidly increased and expanded in southern Venezuela since 2009 [29], particularly in the municipality of Sifontes, state of Bolívar. Changes in land cover patterns resulting from deforestation can promote the emergence of larval habitats for these *Nyssorhynchus* (also known as *Anopheles*) vectors, thereby increasing mosquito abundance [38], vector-host contact [39], survival [40] and consequently transmission risk [41], especially in human settlements located near the forest fringe. Earlier studies in our study area have indicated that the most productive breeding site types for *Ny. albitarsis* s.l. and to a lesser degree for *Ny. darlingi*, are abandoned open lagoons or mining dug-outs left after clearing vegetation [42,43]. Thus, a hypothesis to be tested in future studies would be how forest fragmentation by mining activities may influence both vector species' ecology and vectorial capacity. This ecological mechanism should be addressed regardless of the increase in malaria in the area due to an increased population attracted by the gold mining activities.

The malaria epidemic in Venezuela has been fueled by financial constraints for the procurement of malaria control commodities (such as insecticides, drugs, diagnostic supplies, and mosquito nets) and surveillance activities, and lack of provision and implementation of services [6,44]. In particular, the economic crisis has ignited the informal sector, with illegal gold mining featuring as one of the country's fastest-growing shadow economic activities [45]. As a result, migration within the country has increased towards southeastern Venezuela (Sifontes Municipality) where gold mining activities are clustered and where malaria hotspots have been identified [44]. Highly mobile human populations migrate from different regions of the country to these mining areas in search of economic opportunities. Some of these internal

migrants return back to the region of origin where viable anopheles vector populations exist, reintroducing malaria to areas where autochthonous transmission had been previously eliminated leading to a shift in the epidemiology of this disease [4,5,7]. Concomitantly, the dismantling of the epidemiological surveillance systems at the national level has unintentionally not prevented the reappearance of malaria across the country [7]

The spatial connectedness of malaria foci related to (infected) individuals' travel behavior has been used to identify so-called sources and sinks of *Plasmodium* within a transmission network approach [46]. Sources (originating and exporting cases) and sinks (receiving imported cases) of malaria parasites mostly occur in epidemiological settings when humans frequently travel between locations characterized by substantial heterogeneity in malaria transmission [15,47]. Consequently, we hypothesize that a source-sink malaria parasite (metapopulation) dynamic accounts for the current spatial distribution and persistence of this infection in Venezuela. As we showed, since 2014, malaria has spread from the South and northeastern coastal regions to the lowland central savannas and west Andes piedmont ecoregions where malaria had previously been eliminated, increasing the population at-risk to around 50% compared to 34.4% in 2010. Our results also suggest the importance of malaria linked to mobile illegal gold miners and their role in its spread among the Guiana Shield countries [48–51]. Understanding the spatiotemporal variability between *Plasmodium* and human movement (spatial demography) will help design and implement a strategic malaria control program in Venezuela.

The rapidly increasing malaria burden in Venezuela is affecting neighboring countries, particularly Brazil and Colombia [7]. Migration to Brazil occurs via Route 10, the sole highway that links southern Venezuela and northern Brazil. Unfortunately, this road runs through the hotspots of San Isidro and Dalla Costa. Infected people crossing these borders complicate each country's malaria program by the potential reintroduction of the disease in areas where it had previously been reduced or eliminated; enhancing malaria transmission near border areas; promoting case spillover, and possibly spreading drug resistance alleles across Brazil. Another important risk includes the effects of inadequate treatment that can increase malaria-related mortality. According to the Brazilian Ministry of Health, malaria cases imported from Venezuela into neighboring Roraima State, Brazil have increased from 1,538 (2014) to 4,478 (2018), representing up to 85% of the reported malaria cases of that border state [7,25].

Finally, the spatial distribution of mutations linked to drug-resistant *P. falciparum* in Venezuela remains poorly understood [17]. However, surveillance for resistance markers is now recommended in some Guiana Shield countries such as Guyana and in the mining areas where artemisinin is available for sale and self-treatment is common [52]. Considering that there are multi-drug resistant *Plasmodium falciparum* lineages against common anti-malarial drugs in Venezuela [17,52,53], and the recent report of novel mutations linked to the delay of artemisinin efficacy in Guyana [16], it is possible that multi-drug resistant parasite lineages, that also include artemisinin-response delayed mutations, could emerge in the southern Venezuela hotspots.

The study's main limitations are related to the nature of passive malaria surveillance data and the lack of local demographic information to standardize the analysis of occupational risk. Sub-reporting of malaria incidence is presumed due to a limited diagnostic capacity of the parasite detection in the region and the whole country as a result of the Venezuelan Health crisis. The lack of completeness of the surveillance systems was also a critical issue when we analyzed the 2017 malaria data. Although these facts underestimate the total number of malaria infections, incidence data analyzed still gave us insights about the upward trend of malaria transmission in Venezuela and the localization of the main source (hotpots) of *P. vivax* and *P. falciparum* cases in recent years as suggested by earlier studies [7,13,21]. Similarly, the gold mining as a significant disease social driver and the land-use change (deforestation) as an

important local environmental associated proxy of local malaria increase were such strong patterns that they emerged even with this limited data. The current findings support observations of our previous work [5–7,12,13,17,19,21,22,42, 43]; nevertheless, future studies will be needed to further confirm these patterns.

## Conclusions

We show evidence that gold-mining activities seem to drive malaria hotspots in Venezuela and those high transmission pockets were critical in the surge of malaria observed from 2014 onwards. These gold-mining areas likely make malaria transmission resilient to interventions. In particular, they not only sustain transmission but also can restore it ("rescue effect") after interventions have reduced malaria locally or even achieved local elimination in other areas, as has been observed in Venezuela [15,18]. Transmission in such settings is considered the Gordian knot for achieving malaria elimination in Latin America [5,54–56].

Hotspot-targeted control has been hypothesized as a practical approach to reducing the malaria burden in areas of heterogeneous malaria transmission [18,56]. Thus, a program focused on rapid diagnosis and timely treatment, vector control, and monitoring for drug/insecticide resistance is urgent and essential in these hotspots. Malaria surveillance and reporting have been particularly affected by Venezuela's healthcare crisis since mid-2017 onwards [7]. Hence, to recover and strengthen this surveillance system is a necessary step to control malaria in Venezuela. Increased enforcement of the malaria control program in the rest of Venezuela is pivotal to lower the risks of reintroduction to vulnerable areas. Given the current context, successful control of the ongoing malaria epidemic in Venezuela requires national and regional coordination, as evidenced by the cross-border malaria spillover. Without coordinated international efforts, the progress achieved toward malaria elimination in Latin America over the past 18 years could be easily reversed.

## Supporting information

**S1 Fig. Annual *P. vivax* incidence (Number of confirmed malaria cases/1,000 inhabitants) per parish in Bolívar state (2008–2016), south-eastern Venezuela.** Maps were created with the Q-GIS software (https://www.qgis.org/es/site/)
(TIF)

**S2 Fig. Annual *P. falciparum* incidence (Number of confirmed malaria cases/1,000 inhabitants) per parish in Bolívar state (2008–2016), south-eastern Venezuela.** Maps were created with the Q-GIS software (https://www.qgis.org/es/site/).
(TIF)

**S3 Fig. Significant clusters (hotspots) of annual *Plasmodium vivax* incidence (red-red) in Bolívar state (south-eastern Venezuela), from 2008–2016.** Maps were created with the Q-GIS software (https://www.qgis.org/es/site/)
(TIF)

**S4 Fig. Significant clusters (hotspots) of annual *Plasmodium falciparum* incidence (red-red) in Bolívar state (south-eastern Venezuela), from 2008–2016.** Maps were created with the Q-GIS software (https://www.qgis.org/es/site/).
(TIF)

**S5 Fig.** Age distribution of *P. vivax* (upper panel: A, B) and *P. falciparum* (lower panel: C, D) among malaria patients of San Isidro (left side: A, C) and Dalla Costa (right side: B, D) parishes

(northeastern of Bolivar state) across the study period.
(TIF)

## Acknowledgments

We acknowledge the logistic support (field and laboratory) provided by Angela Martinez, Porfirio Acevedo, Nelson Moncada, Mayida El Souki, Virginia Behm, Manuel Amarista, and Maria A. Oliveira-Miranda. This work is the result of the collaborative work carried out by the research network for vector-borne diseases surveillance & control in Venezuela & Andean region (VeConVen: https://www.vbdvenezuelanetwork.com/).

## Author Contributions

**Conceptualization:** Maria Eugenia Grillet.

**Data curation:** Maria Eugenia Grillet, Juan V. Hernández-Villena.

**Formal analysis:** Maria Eugenia Grillet, Juan V. Hernández-Villena, Maria F. Vincenti-González.

**Funding acquisition:** Maria Eugenia Grillet, Jorge E. Moreno, Jan E. Conn.

**Investigation:** Maria Eugenia Grillet, Jorge E. Moreno, Juan V. Hernández-Villena, Jan E. Conn.

**Methodology:** Maria Eugenia Grillet, Juan V. Hernández-Villena, Maria F. Vincenti-González.

**Project administration:** Maria Eugenia Grillet, Jan E. Conn.

**Resources:** Maria Eugenia Grillet, Jorge E. Moreno, Oscar Noya, Adriana Tami, Alberto Paniz-Mondolfi, Martin Llewellyn, Rachel Lowe, Ananías A. Escalante, Jan E. Conn.

**Software:** Maria Eugenia Grillet, Juan V. Hernández-Villena, Maria F. Vincenti-González.

**Supervision:** Maria Eugenia Grillet.

**Validation:** Maria Eugenia Grillet, Jorge E. Moreno.

**Visualization:** Juan V. Hernández-Villena, Maria F. Vincenti-González.

**Writing – original draft:** Maria Eugenia Grillet, Ananías A. Escalante, Jan E. Conn.

**Writing – review & editing:** Maria Eugenia Grillet, Jorge E. Moreno, Oscar Noya, Adriana Tami, Alberto Paniz-Mondolfi, Martin Llewellyn, Rachel Lowe, Ananías A. Escalante, Jan E. Conn.

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
