## [Decision Letter · Decision Letter 0]

12 May 2020

Dear Prof. Grillet,

Thank you very much for submitting your manuscript "Malaria in Southern Venezuela: The Hottest Hotspot in Latin America" for consideration at PLOS Neglected Tropical Diseases. As with all papers reviewed by the journal, your manuscript was reviewed by members of the editorial board and by several independent reviewers. In light of the reviews (below this email), we would like to invite the resubmission of a significantly-revised version that takes into account the reviewers' comments. 

We cannot make any decision about publication until we have seen the revised manuscript and your response to the reviewers' comments. Your revised manuscript is also likely to be sent to reviewers for further evaluation.

Sincerely,

Rhoel Ramos Dinglasan

Associate Editor

Marcelo Ferreira

Deputy Editor

Reviewer's Responses to Questions

**Key Review Criteria Required for Acceptance?**

**Methods**

-Are the objectives of the study clearly articulated with a clear testable hypothesis stated?

-Is the study design appropriate to address the stated objectives?

-Is the population clearly described and appropriate for the hypothesis being tested?

-Is the sample size sufficient to ensure adequate power to address the hypothesis being tested?

-Were correct statistical analysis used to support conclusions?

-Are there concerns about ethical or regulatory requirements being met?

Reviewer #1: I believe that the methods employed in this article could be substantially improved. Specific comments are provided in the attached document.

Reviewer #2: The objectives are clearly outlined. However, in reading, there is some confusion as to what starting points represent hard data, and what are modeled estimates. The methods section could be improved with more clarity addressing this. Given the current situation there in Venezuela, for example, what are the pitfalls of using either VMoH or even PAHO data? No concerns in re ethics or regulatory requirements. More generally, the main objective seems to be to identify the hotspots, yet there is a lot of text in the paper that does not really focus the reader's attention on this.

**Results**

-Does the analysis presented match the analysis plan?

-Are the results clearly and completely presented?

-Are the figures (Tables, Images) of sufficient quality for clarity?

Reviewer #1: I do not think that the results are clearly presented and the figures could be substantially improved. Specific comments are provided in the attached document.

Reviewer #2: The image quality in Figure 1 is poor. In figures 3 and 4, since no denominator is indicated, one assumes that what is illustrated is total number of cases; this is perhaps less useful to those not familiar with the population distribution in Venezuela than if the data were converted to incidence RATE. Since the pattern stays pretty similar from year to year, would a single consensus map be more useful for the reader, with this more granular imaging relegated to supplementary figures? Similarly, figures 5 and 6, although highly illustrative, are also highly repetitive, and could be delegated to supplementary figures, with a representative year, perhaps the most recent, having a greater impact on the reader, with the additional mapping available for perusal if desired.

**Conclusions**

-Are the conclusions supported by the data presented?

-Are the limitations of analysis clearly described?

-Do the authors discuss how these data can be helpful to advance our understanding of the topic under study?

-Is public health relevance addressed?

Reviewer #1: Some of the conclusions are not well supported by the data they present. Specific comments are provided in the attached document.

Reviewer #2: The modeling itself appears sound. Some of the material in the discussion section reads more like background, and could perhaps be consolidated with that. Instead, a more thorough discussion of the other assumptions (including estimating the reliability of the starting data) would be most helpful. The discussion of r nought is quite good. However, more discussion on the population migrations and their changing patterns over time would tie the paper together better as well. What about urbanization?

**Editorial and Data Presentation Modifications?**

Reviewer #1: Specific comments are provided in the attached document.

Reviewer #2: 1. Some of the methods/explanation of terms could be put in supplemental material – it's very educational, but it contributes to the MS length.

2. Some statements need to be softened. A good example is on line 239, where it might be better to either state that the decreasing vegetation is ASSUMED to be attributable to mining activities, or include a citation for this if you have it. 

Very minor points are: Line 10; I think you mean to say "As in other malaria-endemic countries in SA, PV dominates, accounting for..." (I doubt that the numbers are exactly the same in all those other countries, which is what you are currently saying!)

line 37: I think you mean "resistance" rather than "resistant" (noun versus adjective)

**Summary and General Comments**

Reviewer #1: (No Response)

Reviewer #2: Overall, the paper is a bit long for the content. There is repetition and excessive attention to some background concepts. Consider what is actually relevant to your methods and results, and, as mentioned above, what helps explain the identification of “hotspots”. For example, the mosquito information given does not adequately support the rather strong statement on lines 231-3; this statement should refer to your results being reported, not possible other work unless referenced. The climatic information provided, although it provides color, also seems more suited to another manuscript in which climactic factors are more addressed.

PLOS authors have the option to publish the peer review history of their article (what does this mean?). If published, this will include your full peer review and any attached files.

Reviewer #1: No

Reviewer #2: No
---

## [Decision Letter · Decision Letter 1]

16 Sep 2020

Dear Prof. Grillet,

Thank you very much for submitting your manuscript "Malaria in Southern Venezuela: The Hottest Hotspot in Latin America" for consideration at PLOS Neglected Tropical Diseases. As with all papers reviewed by the journal, your manuscript was reviewed by members of the editorial board and by several independent reviewers. The reviewers appreciated the attention to an important topic. Based on the reviews, we are likely to accept this manuscript for publication, providing that you modify the manuscript according to the review recommendations. 

Sincerely,

Rhoel Ramos Dinglasan

Associate Editor

Marcelo Ferreira

Deputy Editor

Reviewer's Responses to Questions

**Key Review Criteria Required for Acceptance?**

**Methods**

-Are the objectives of the study clearly articulated with a clear testable hypothesis stated?

-Is the study design appropriate to address the stated objectives?

-Is the population clearly described and appropriate for the hypothesis being tested?

-Is the sample size sufficient to ensure adequate power to address the hypothesis being tested?

-Were correct statistical analysis used to support conclusions?

-Are there concerns about ethical or regulatory requirements being met?

Reviewer #2: The manuscript has been considerably tightened and clarified.

Reviewer #3: (No Response)

Reviewer #4: -the goals of the paper are outlined and the design and data are appropriate for these goals

-the population and data are well described

-the analyses are appropriate - there is one analysis that could be more robust (see summary comments)

-there are no concerns about ethical requirements

-lines 140-145: Can you explain this analysis in more detail? Were the expected frequencies based on each parish?

**Results**

-Does the analysis presented match the analysis plan?

-Are the results clearly and completely presented?

-Are the figures (Tables, Images) of sufficient quality for clarity?

Reviewer #2: The manuscript has been considerably tightened and clarified.

Reviewer #3: (No Response)

Reviewer #4: -There are a few analysis results that were not described in the methods section (see comments below)

-The results are clearly presented. 

-The maps and figures are very nice. In Figure 1, the name label for the civil parish overlaps the location of the capitol city – can this be changed? Also in Figure 1, both the stare and the triangle are labeled at capital cities. For Figures 3 and 4, it does not seem that the civil parish labels can be read; it might be better to remove these names. 

-line 173: Is this p-value the result of an analysis for trend? This analysis should be described in the methods section. 

-lines 174-178: “Bolivar contributed … southeastern region.” This sentence is complicated/confusing – do you mean that more than half of the cases in Venezuela were from Bolivar during the time period?

-lines 179, 181, 182: The descriptive analyses (mean ratio calculations, trend tests) should also be in the methods section

-Figure 6: The lines in the figure clearly represent some analysis (correlation or regression) between the data, but the analysis is not described in the methods section

-Figure 7: What is the unit of measurement? Are these incidence rates?

**Conclusions**

-Are the conclusions supported by the data presented?

-Are the limitations of analysis clearly described?

-Do the authors discuss how these data can be helpful to advance our understanding of the topic under study?

-Is public health relevance addressed?

Reviewer #2: The manuscript has been considerably tightened and clarified.

Reviewer #3: (No Response)

Reviewer #4: -The conclusions are mostly supported - the occupational link would be more convincing with a revised analysis (see summary comments)

-There is no discussion of analysis limitations

-The authors provide an excellent discussion of how these data/analyses could inform public health policy and malaria surveillance/control

-The public health relevance is extremely clear

-line 298-302: “Thus, a … mining activities”. This sentence is complicated/too long. I suggest you split it into two sentences. 

-line 313-314: Can you clarify – these migrants reintroduce malaria parasites when they return to other parts of Venezuela 

from the mines? Do these workers come to the gold mines to work and then return home (or make multiple trips)?

-line 347: What is SP?

**Editorial and Data Presentation Modifications?**

Reviewer #2: Line 3 ‘especially so’ would read better than ‘even so’.

Line 23 comma after ‘healthcare system’

Line 44 comma after ‘neighboring countries’

Line 47 movements is plural so it should be “serve”

Line 76 comma after both ‘Venezuala’ and ‘America’

Line 281, comma after [37]

Line 285 meaning of this sentence is not entirely clear; is the case clustering or the deforestation the consequence of gold mining activity? Please rewrite this sentence for clarity. 

Line 290 should discuss that the perpetrators of illegal gold mining would be expected to be largely young males, so that this association might be related to transmission exposure rather than any kind of independent “other”. 

Also line 290, the sentence beginning with ‘Illegal’ seems like the start of a new paragraph, since it signals a switch to discussion of the vector.

Line 349 comma after ‘strains’ 

Line 350 the word ‘response’ should be inserted after ‘artemesinin’, perhaps with a hyphen linking the two.

Figure 1; the labeling of the individual region names, and of the color key, is not publication quality

Figures 3 and 4; as for Figure 1

Figure 5; the key is not as informative as the legend. A preferable key would say ‘localities with <1000 cases’ etc. 

Figure 7; neither the figure itself nor the legend indicates the units, presumably cases, but per what denominator?

Reviewer #3: (No Response)

Reviewer #4: -Some word choices are unusual or there are places where the sentences could be clearer and there are minor typos (see minor comments below for these suggestions). Overall, the manuscript is readable and the organization makes sense. 

-line 2: change “the last” to “recent”

-line 3: change “even so” to “particularly”

-line 5: remove “largely” (it is redundant)

-line 46: change “movements” to “movement”

-line 61: change “malaria resilience” to “the resilience of malaria”

-line 76: put a comma after “Venezuela”

-lines 76, 80, 81: format “Km2” as “km²” 

-lines 90, 91: change “exhibits lacks of seasonality” to “does not exhibit a seasonal pattern”

-lines 124-125: change “and then” to “but have been”

-line 128: health should be capitalized

-line 150: remove “The”

-line 151: change “the last” to “recent”

-line 199: remove the comma

-line 201: remove “also” 

-Table 1: the parish is labeled “San Ignacio” but the text says “San Isidro”

-line 240: define “Ha”

-line 280: change “sum factors” to “sum of factors”

**Summary and General Comments**

Reviewer #2: There is still not a short discussion of the strengths and pitfalls of reliance on the source data; please include a sentence or two, and perhaps a reference, on the utility and reliability of this data.

Reviewer #3: Here is my full review:

“Malaria in Southern Venezuela: The Hottest Hotspot in Latin America”

This is an informative study of the current deteriorating malaria situation in Venezuala. The study is very well done and clear. A strength is that the authors use available data from throughout the country to show hotspot zones of high transmission. They show how goldmining is associated with malaria cases and how deforestation is a cause of increased malaria incidence. This paper will be of high interest to readership as it lays out public health challenges facing Venezuela and the region.

The revision adequately addresses all the reviewer’s comments. The manuscript is now acceptable. The authors do not need to revise again.

A point I really like in the manuscript is the reference 9 citation of Gabaldon. Professor Gabaldon was a real pioneer in malaria control. He and his colleagues moved Venezuela close to malaria elimination. At one time Venezuela led all other South American countries in the fight against malaria. Now everything has collapsed but surely Gabaldon will long be remembered!

Reviewer #4: This manuscript provides an overview of malaria epidemiology in Venezuela and assesses the presence of hotspots for the disease over 2007-2017. The authors examine these hotspots and their link to gold mining and deforestation. They also discuss country-wide increases and inadequate malaria control programs in the country. The authors call for control programs using the information gained from this analysis of spatial epidemiology. The manuscript is interesting and assesses the spatial epidemiology of malaria with multiple analyses and from multiple angles. The discussion on gold mining, deforestation, and malaria is especially thoughtful and interesting. 

-For the analysis of occupations, gender, and age structure among cases in hotspots (Tables 1 and 2), there needs to be additional detail in the methods section. It seems that the analysis is comparing the distribution of cases among categories within each parish. The link between occupation, gender, age, and malaria infection would be more convincing if the authors performed something like an indirect standardization. It is important to account for the underlying populations in these parishes – if 60% of the people living in San Isidro work in gold mining, we would not be surprised if 60% of people with malaria are gold miners – maybe malaria and gold mining are unrelated and San Isidro just has lots of people who work in mining. If malaria is truly related to gold mining, we first must account for San Isidro having many gold miners living there. If there are more gold miners with malaria than we would expect (given the high number of miners in the parish), then we can conclude that there is an overrepresentation of gold miners among malaria cases. 

-Can you provide some background information (a couple sentences) on the link between deforestation and gold mining and malaria in the introduction/background section?

-Can you discuss the limitations of only having data from EW 1-21 in 2017 for the spatial prediction maps?

PLOS authors have the option to publish the peer review history of their article (what does this mean?). If published, this will include your full peer review and any attached files.

Reviewer #2: Yes: V. Ann Stewart

Reviewer #3: No

Reviewer #4: Yes: Rachel Sippy
---

## [Editor Report · Decision Letter 2]

25 Nov 2020

Dear Prof. Grillet,

We are pleased to inform you that your manuscript 'Malaria in Southern Venezuela: The Hottest Hotspot in Latin America' has been provisionally accepted for publication in PLOS Neglected Tropical Diseases.

Best regards,

Rhoel Ramos Dinglasan

Associate Editor

Marcelo Ferreira

Deputy Editor

---

## [Editor Report · Acceptance letter]

20 Jan 2021

Dear Prof. Grillet,

We are delighted to inform you that your manuscript, "Malaria in Southern Venezuela: The Hottest Hotspot in Latin America," has been formally accepted for publication in PLOS Neglected Tropical Diseases.

Best regards,

Shaden Kamhawi

co-Editor-in-Chief

Paul Brindley

co-Editor-in-Chief
